# Resource Management for Collaborative 5G-NR-V2X RSUs to Enhance V2I/N Link Reliability

**DOI:** 10.3390/s23083989

**Published:** 2023-04-14

**Authors:** SangHoon An, KyungHi Chang

**Affiliations:** Department of Electrical and Computer Engineering, Inha University, Incheon 22212, Republic of Korea; tkdgnsdldb@gmail.com

**Keywords:** 5G-NR-V2X, RSU, cooperative resource management, cell range expansion, 3D beamforming

## Abstract

In the development of autonomous driving technology, 5G-NR vehicle-to-everything (V2X) technology is a key technology that enhances safety and enables effective management of traffic information. Road-side units (RSUs) in 5G-NR V2X provide nearby vehicles with information and exchange traffic, and safety information with future autonomous vehicles, enhancing traffic safety and efficiency. This paper proposes a communication system for vehicle networks based on a 5G cellular network with RSUs consisting of the base station (BS) and user equipment (UE), and validates the system performance when providing services from different RSUs. The proposed approach maximizes the utilization of the entire network and ensures the reliability of V2I/V2N links between vehicles and each RSU. It also minimizes the shadowing area in the 5G-NR V2X environment, and maximizes the average throughput of vehicles through collaborative access between BS- and UE-type RSUs. The paper applies various resource management techniques, such as dynamic inter-cell interference coordination (ICIC), coordinated scheduling coordinated multi-point (CS-CoMP), cell range extension (CRE), and 3D beamforming, to achieve high reliability requirements. Simulation results demonstrate improved performance in outage probability, reduced shadowing area, and increased reliability through decreased interference and increased average throughput when collaborating with BS- and UE-type RSUs simultaneously.

## 1. Introduction

With the recent development of autonomous driving, interest in V2X technology has been increasing. Researchers have explored V2X technology to maximize traffic safety and road efficiency by comprehensively utilizing the state of vehicles in use and other vehicles on the road and road conditions. The telecommunications and automobile industries are requesting the transmission of more detailed messages to support V2X services. These requirements necessitate meeting key performance indicators (KPIs), such as data speed, reliability, latency, communication range, and movement speed through the development of V2X applications [1]. In this sense, 5G-New Radio (NR) technology [2] provides a flexible design and higher energy efficiency to support lower latency, higher throughput, broader coverage, and more reliable services, thus meeting the required KPIs. In particular, the International Telecommunication Union—Radio (ITU-R) introduces three groups of enhanced usage scenarios covering 5G-NR characteristics, which require a more flexible, reliable, and secure service delivery than ever before [3]. In three scenarios, enhanced mobile broadband (eMBB), ultra-reliable and low-latency communications (URLLC), and machine-type communications (mMTCs), eMBB aims to provide faster data transmission using a higher frequency bandwidth and more antennas to handle services that require large transmission in dense locations. In particular, it aims to provide high speed in areas with strong signals near base stations and areas with weak signals at the edge of cells. URLLC is a scenario that satisfies the requirements for features, such as throughput, latency, and availability, and enables services that require real-time response speed for autonomous vehicles by communicating surrounding traffic conditions [4]. Finally, mMTC enables large-scale communication services suitable for low data rates and low-cost devices for large numbers of devices based on high reliability and low latency. To solve these diverse and demanding requirements, 3GPP NR V2X supports and studies more advanced use cases, such as vehicle platooning, extended sensors, advanced driving, and remote driving, as classified in [5].

As mentioned above, the high reliability requirements of V2X communication pose significant technical challenges, and V2X communication enables desired capacity, high throughput, and short latency using vehicle-to-vehicle (V2V), vehicle-to-pedestrian (V2P), vehicle-to-infrastructure (V2I), and vehicle-to-network (V2N) technologies. In recent years, both industry and academia have been continuously researching to meet the demands of low latency and high reliability in vehicle networks, and several surveys and tutorials have been published. Although many studies have been conducted using a combination of PC5 and Uu interface-based V2X communication, research and details on the use of RSUs of different types supporting various functions have not been covered. A single type of BS/UE RSU is not sufficient to support V2X services when communicating with vehicle terminals. Therefore, this study proposes a method for utilizing cooperative BS/UE-type RSUs to improve system performance by reducing interference between receivers and maximizing the average throughput of vehicles for the prompt and stable delivery of received messages. The proposed cooperative utilization of BS/UE-type RSUs in this paper supports high reliability and throughput in a 5G-NR V2X environment and guarantees strict quality of service (QoS) requirements for V2I/N communication by utilizing links for V2X communication when traffic loads increase in urban environments.

Based on such V2X-related scenarios, the performance analysis of reliability enhancement, reduced shadowing areas in dense urban environments, and interference between vehicle receptions are presented together with the average throughput when implementing a system through mutual cooperation between BS/UE-type RSUs. Additionally, we introduce various techniques through different heterogeneous parameters that can be applied when utilizing each RSU to maximize the average throughput. Our main contribution in this paper is as follows.

This paper emphasizes the characteristics of RSUs used in V2X based on 3GPP technical reports to better understand the functionality of V2X within the 5G NR system, and discusses 5G-V2I/V2N modeling in dense urban scenarios. We design a cellular system model in a dense urban environment using the SLS methodology proposed by 3GPP for V2X performance comparison based on rel. 16. The methodology provides standards for the distribution of each RSU type, road composition, and vehicle arrangement. In addition, it incorporates standardized criteria such as antenna models, channel models, and PHY layer models.We propose a hybrid RSU approach as a cooperative system between BS/UE-type RSUs in the designed system environment and minimize shadow areas through an RSU allocation method based on SINR received at the V2X application server. The proposed Hybrid RSU approach not only improves the overall received SINR of the VUEs in the simulation, but also shows performance improvements in received interference and average throughput. Moreover, it provides insights into the impact of using different RSU approaches in cooperation on the overall system performance.The proposed Hybrid RSU technology approach contributes to higher system capacity or better power usage in direct communication between vehicle terminals and RSUs through resource collaboration management, CRE, and 3D beamforming technology. The applied cooperative resource allocation techniques, dynamic ICIC and CS-CoMP are effective methods for mitigating Co-Channel Interference (CCI) and improving the channel situation of cell edge users in dense urban environments. Furthermore, the utilization of 3D beamforming technology is effective in meeting the safety, reliability, and high-quality driving requirements of the V2X environment. Realistic antenna radiation patterns are characterized and applied to evaluate performance in 5G cellular scenarios.Simulation results focus on analyzing VUE interference, outage probability, and average throughput values by the proposed Hybrid RSU approach. The simulation results demonstrate that the proposed scheme significantly improves the channel quality and overall system reliability.

The structure of this paper is as follows: Section 2 reviews prior research and proposed techniques to maximize RSU performance in V2X communication, and Section 3 presents a system model that constitutes a simulation environment. Section 4 introduces how BS/UE-type RSUs collaborate for V2X communication and evaluates performance using system-level simulation (SLS) when applying the techniques proposed in Section 5. Finally, Section 6 provides our conclusion and recommendations.

## 2. Related Works

This section introduces the mutual cooperation scheme conceived in the V2V and V2I/N communication-based systems using the RSU, composed of the existing BS- and UE types. We then review prior studies to utilize RSUs and the proposed techniques to maximize the performance of each RSU.

The current 3rd Generation Partnership Project (3GPP) and 5G Automotive Association (5GAA) standards define wireless interface Uu and PC5 interface-based communications for transmitting and receiving V2X messages. The Uu interface-based communication is a cellular network-enabled support V2I/N that communicates between a BS-type RSU and vehicle users (VUEs), providing high reliability and information transmission rates. On the other hand, the PC5 interface-based communication connects with different VUEs and UE-type RSUs, reduces latency, and offers fast data transmission in areas that do not belong to the network area. In a way that uses both Uu and PC5 interfaces, VUE communicates with UE-type RSUs over the PC5 interface and other nearby VUEs to send and receive V2X messages, and UE-type RSUs communicate with V2X servers via the Uu interface to manage the communication of V2X messages beyond the required PC5 communication range. This mixed-use PC5 and Uu interface-based V2X communication can reduce latency with RSUs to communicate with local V2X servers, because there is no need for Multimedia Broadcast or Multicast Service (MBMS) on the downlink [6,7].

Research on V2X communication using a combination of PC5 and Uu interfaces has been ongoing recently, as evidenced by [8,9,10]. These studies have proposed algorithms to ensure the reliability of V2V links, while maximizing the total and minimum ergodic capacities of V2I links, considering the sharing of the spectrum between multiple V2V links and V2I links. Specifically, each resource block can be accessed by different V2I or V2V links, and each vehicle link can utilize different resource blocks based on the assumption that fading information is the only basis for the resource allocation system. As a result, the optimal resource allocation problem has been considered and a power optimization mechanism has been implemented. Furthermore, studies such as [11,12] have formulated the maximum applicability range for hybrid RSU deployment and management to achieve maximum performance in a restricted RSU placement, while mitigating the gap between strict V2X requirements and limited available resources. Researchers have investigated communication reliability, reduced latency, and efficient power allocation through such studies on RSUs, and research on hybrid RSU deployment has been ongoing.

In addition, various techniques and configurations have been applied and tested to enhance the V2I/N link reliability in ongoing research. The applied techniques and configurations demonstrate significant improvements in the vehicle–terminal interference and average throughput, aim to mitigate interference between BS-type RSUs, and improve communication reliability through two cooperative interference management systems, Dynamic-ICIC and CS-CoMP, as presented in [13]. In addition, [14] proposed QoS-based tuning scheduling and a hybrid spectrum access scheme for downlink CoMP transmission in a heterogeneous network environment. The proposed method dynamically reduces the power demand of cells considering adjacent cell interference, and balances the load of macro cells by switching small cells to the hybrid access mode. This method enables both CS-CoMP, inter-cell interference, and unnecessary muting of base stations simultaneously. Moreover, [15] combines horizontal beam pattern adaptation applied to 3D beamforming and MIMO (Multi-Input Multiple-Output) with vertical antenna pattern adaptation. This application processes fully dynamic antenna pattern adaptation designated for each resource block and UE to enable the practical use of 3D beamforming. This paper verifies a decrease in shadow areas and improvement of communication reliability in the system by applying the various techniques and configurations introduced above.

## 3. System Model of Cellular V2X Network

This study implements a 3GPP general deployment scenario that provides networks, including V2V, V2I/N, and V2P services [16]. The study by [14] presents a methodology for evaluating new features required to operate LTE-based V2X services, and uses it to compare performance when applying various technologies [17]. On the other hand, [15] studies the evaluation methodology of new V2X use cases for LTE and NR, comparing the performance of various technology options for new 5G V2X use cases [18]. Further, [16] investigates NR-V2X to address the improvement of the Uu interface to support improved V2X use in NR-V2X, and the allocation/configuration of Uu interface-based sidelink (SL) resources by LTE and NR [19]. This section covers system setup, including network layout, antenna model, channel model, and physical layer, to achieve requirements in 5G NR environments by applying the simulation methodology of the mentioned 3GPP documents. In this study, we perform simulations using link-to-system (L2S) mapping through the utilization of link-level simulator (LLS) and system-level simulator (SLS) based on MATLAB. We implement our own SLS based on MATLAB using the system model presented in Section 3.1, Section 3.2, Section 3.3; and for LLS, we utilize the Vienna 5G LL Simulator, an open-source simulator based on MATLAB, to model link-level performance according to the 5G LDPC codes design approach described in Section 3.4, and apply it to the system.

### 3.1. 5G-NR-V2X Network Layouts

In this study, we construct the cellular network considered for evaluation using the SLS of 5G-NR-V2X under the Manhattan urban environment of 3GPP technical documents [17,18,19] mentioned in the previous section. The Manhattan environment for evaluation in a 5G-NR-V2X is a macro model in an urban environment that is continuously mentioned in the 3GPP document [17,18,20]. The proposed urban environment simulation area is a one-tier network, with a horizontal length of 250 m and a vertical length of 433 m between the road grids between the intersections. Therefore, the minimum size in the simulation is 750 m × 1299 m. In the case of roads for VUE circulation, it is assumed that each block edge has a two-lane road with a total width of 3.5 m per lane, making the total road width 7 m. The 3GPP standard defines fixed infrastructure entities that support V2X applications that can exchange messages with other entities, such as RSUs [16]. Further, RSUs have two definitions, BS-type RSUs and UE-type RSUs, deployed using wrap-around methods in the present network layout. In the case of BS-type RSUs, seven BS-type RSUs comprise 7 × 3 cells that satisfy three sectors per site, and seven BS-type RSUs in the simulation satisfy 500 m of the inter-site distance (ISD) between BS-type RSUs. In the case of UE-type, RSUs are placed in the center of the intersection in an urban environment, and a total of 16 UE-type RSUs configure within the present simulation.

We distribute the simulation’s VUE on the road according to the spatial Poisson process, and the vehicle density distribute on each road depends on the vehicle speed assumption. The spatial Poisson process used is a method of distributing points along the road based on the occurrence of point events within a specific area of interest [21]. All assumptions in the simulation are based on the methodology presented in the 3GPP technical report, which assumes vehicles with the same speed and density. In dense urban environments, the VUE has an antenna height of 1.5 m and an absolute speed of VUE is set to 60 km/h, and to define vehicle density, the average vehicle-to-vehicle distance in the same lane is set to 2.5 s × absolute vehicle speed. The 1D Poisson point process (PPP) method is applied to model the positions of vehicles in the simulation, using the formula implemented and in accordance with Equation (1).
(1)$$P\left(N\left(D\right)=K\right)=\frac{e^{-\lambda D}{\left(\lambda D\right)}^{k}}{k!}\;\mathrm{for}\;k=0,1,2,3,\;\ldots ,\;N$$

Here, parameter *λ* represents the point density of each lane, *D* represents the length of each lane, and *N*(*D*) represents the number of vehicles located in each lane. Once the vehicle density parameter *λ* for each lane is determined, Equation (2) is applied to evenly distribute the points obtained by the PPP onto each lane in the simulation.
(2)$$\overset{N+1}{\underset{j=1}{\prod }}U_{j}<e^{-\lambda }$$

In this context, each *U_j_* value represents an independent random variable uniformly distributed between 0 and 1, while *N* denotes the number of vehicles in each lane, obtained from Equation (2). In addition, it is assumed that all VUEs go straight at the intersection without changing direction, and that all potential collisions at the intersection are ignored. In order to eliminate potential simulation errors due to a single density and velocity in these assumptions, we utilized the simulation results as an ensemble average, which is the average of all possible results that can be generated by a random process.

We use a network configuration that employs both BS-type and UE-type RSUs in the SLS design, as shown in Figure 1b. In Figure 1b, the blue square represents the BS-type RSU, the green circular dot represents the UE-type RSU, and the red square represents the distribution of each VUE.

### 3.2. Antenna Model

Antenna models are the most influential factors in performance evaluation, and the system model configures antenna models of BS-type RSUs and UE-type RSUs differently [15,17]. In the case of an antenna of the BS-type RSU, it is a configuration of an antenna that satisfies three sectors in a hexagonal grid and has a height of 25 m. The number of antenna elements of all panels of the BS-type RSU antenna is up to 256 TX/RX antenna elements. The array model is a uniform rectangular panel array represented by *M* (number of antenna elements in each row), *N* (number of antenna elements in each row), *P* (polarized type single-polarized or dual-parallelized), M_g_ (number of antenna elements in each row), and *N_g_* (number of antenna elements in each column). In 3GPP, the configuration of the antenna array of the BS-type RSU is (*M*, *N*, *P*, *M_g_*, *N_g_*) = (8, 8, 2, 1, 1). The antenna panel configuration includes a uniform interval that satisfies (*d_H_* (distance between horizontal arrays), *d_V_* (distance between vertical arrays)) = (0.5, 0.8)*λ*. In addition, the UE-type RSU has a different antenna configuration than the BS-type RSU. which height of the UE-type RSU is 5 m, and the number of antenna elements on all panels is up to 8 TX/RX antenna element which consists of (1, 2, 2, 1, 1). The interval between the antenna arrays has a uniform interval satisfying (*d_H_*, *d_V_*) = (0.5, 0.8)*λ*. In the antenna configuration, a larger number of antenna elements increase the power received from the VUE, enabling narrower beams and higher accuracy [22]. Section 5.3 considers 3D beamforming proposed by 3GPP in BS-type RSUs, and identifies several limitations related to the range of adjustments caused by vertical cell geometry. It also introduces 3D beamforming, which combines the horizontal beam pattern adaptation and vertical antenna pattern adaptation with vertical pattern adaptation, maximizing the VUE throughput performance according to the increase in the receiving SINR of the vehicle terminal.

### 3.3. Channel Model

The propagation loss for the transmission link at each RSU considers pathloss, shadowing, and fast fading. The main propagation loss model for the two proposed RSU types is implemented and used according to (3) [13].
G = Antenna_Gain_ − PathLoss − Shadowing − Fading (3)

Table 1 outlines the applied assumptions about the channel model between VUE and RSU, and the parameters are considered by applying the 3GPP TR SLS methodology accordingly [17,18,19].

The pathloss model used in this study assumes the channel model between BS-type RSU and VUE according to the 3GPP document [17], and the WINNER model for V2V channel assumption between UE-type RSU and VUE. For the BS-type path loss model, only LOS is applied, and for the WINNER + B1 Manhattan, grid layout urban microcell scenario used for the UE-type RSU, the antenna heights of VUE and RSU are assumed to be lower than the top of surrounding buildings, and LOS and NLOS cases are defined and calculated for all positions leading to the RSU [22]. Additionally, to form the shadowing correlation between RSUs, shadowing was applied to the BS-type RSUs using shadowing correlation coefficient 0.5 for inter-site shadowing and 1.0 for inter-sector shadowing.

In the case of fading, the 3GPP TR utilizes the ITU-R Urban Micro-Clustered Delay Line Models (CDL) with large, fixed parameters for NLOS [23] in the propagation model of the UE-type RSU, while the propagation model for the BS-type RSU utilizes the 3GPP spatial channel model (SCM) NLOS to calculate the radio gain.

### 3.4. Abstraction of the Physical Layer

In this section, we obtain the packet error rate (PER) by computing VUE throughput using specific modulation and coding levels for abstraction in the physical layer (PHY). 5G-NR uses LDPC coding rather than turbo coding used in LTE. The 15-channel quality indicator (CQI) indexes using the modulation coding scheme (MCS) that provides up to 256QAM, achieving different spectral efficiencies at various MCS levels, represent 5G-NR. In these PHY calculations, we must assume that urban environment VUE reception signals have a perfect time and frequency synchronization; otherwise, obtaining PERs through system-level simulation (SLS) and performing measurements considering all transmission and reception architectures for signal processing is expensive and time-consuming. Therefore, it is necessary to simplify and apply limited assumptions and systems for simulation analysis, using link-level simulation (LLS) to evaluate analysis results under realistic conditions. For LLS utilization, we implement and apply the LLS curve of the additive white Gaussian noise (AWGN) channel, using the Vienna 5G Link Level Simulator [24]. The simulation results shown in Figure 2. Compare with the simulation results in LTE, 5G NR, and the analysis performed on the AWGN channel with a CQI value of 1 to 15 [25]. Simulation results provide calibration values for effective SINR mapping in feedback calculations, along with comparison of LLS performance [24]. Therefore, we conduct link-to-system (L2S) mapping based on the results using mutual information effective SNR mapping (MIESM). Effective SNR mapping (ESM) algorithms provide the important link–layer abstraction for SLS. Thus, for MIESM application in this study, we compute based on nonlinear mapping relationships from SINR to mutual information to compute an effective SINR, utilizing (4) and (5) to provide a higher accuracy than other ESM methods.
(4)$$SINR_{eff}=\beta \cdot I^{-1}\left(\frac{1}{N}\overset{N}{\underset{n=1}{\sum }}I\left(\frac{SINR_{n}}{\beta }\right)\right)$$
(5)$$I_{mp}\left(x\right)=m_{p}-E_{y}\left\{\frac{1}{2^{mp}}\overset{mp}{\underset{i=1}{\sum }}\overset{1}{\underset{b=0}{\sum }}\underset{z\in X_{b}^{i}}{\sum }log\frac{\sum_{\tilde{x}\in X}\exp\left(A\right)}{\sum_{\tilde{x}\in X_{b}^{i}}\exp\left(A\right)}\right\}\;A=-{\left|Y-\sqrt{x∕\beta }\left(\tilde{x}-z\right)\right|}^{2}$$

Here, *β* is the calibration coefficient selected to minimize the mean square error between the effective SNR derived from the Rayleigh channel and the fixed SNR [26], and such measurements are based on the mutual information function, *I*(*SINR_n_*). Furthermore, *I_mp_* in Equation (5) represents the number of bits per symbol for the chosen modulation scheme, where *X* is the symbol set and *Y* is a zero-mean complex Gaussian variable with unit variance [27].

## 4. User Association Scheme for Collaborative RSUs

In this section, we propose a hybrid RSU method, which is the mutual cooperation scheme of BS/UE-type RSUs for V2X communication when deploying UE-type RSUs in the coverage area of urban Macro BS-type RSUs in a 5G-NR-V2X network. The proposed hybrid RSU method efficiently allocates V2X messages using BS/UE-type RSUs, to secure efficient user services. The hybrid RSU method minimizes shaded areas generated in 5G-NR-V2X environments, to use the data more efficiently and reliably for real-time changing road and traffic environments. For simulation evaluation, we assume a directional antenna-based cellular network, and a single antenna VUE can be adjusted vertically and horizontally through feedback for the backhaul link and signal delay control in the simulation. Additionally, we assume a cellular network that is always activated in the downlink direction for all sessions when using the hybrid RSU method, and analyze the impact of the proposed hybrid RSU method. In this case, vehicle terminals access the wireless channel to communicate with each RSU, and vehicles accessing the wireless channel undergo resource allocation through a proportional fair scheduler [28]. The scheduler performs resource allocation based on CQI estimated from the received signals. The received signal of each vehicle terminal and the estimated CQI for each vehicle are used to calculate the SINR for all PRBs. The calculated SINR value is used to select the modulation and coding rate MCS system for each vehicle terminal, and affects the average processing throughput value of the vehicle terminal.

### 4.1. Calculating the SINR UE-Tye RSU and BS-Type RSU Channel Model

Each RSU type configures in a cellular multi-tier network consisting of different transmission power and other parameters. The VUE checks the RSU with the highest received SINR at the V2X application server and selects the RSU to serve the VUE. Based on the cellular V2X network presented earlier, we check the shaded area for each RSU type. In addition, we analyze the simulation results using the mutual cooperation of each RSU type, minimizing the shaded area. We conduct the simulation through SLS analysis implemented in the 5G-NR-V2X environment. After configuring the V2X environment considering the network layout (urban case), we evaluate the VUE distribution and mobility model, RSU distribution, channel model, traffic model, vehicle terminal interruption probability, reception interference, and average throughput. The received SINR, according to the RSU type of each vehicle, represents the shaded area within the simulation environment; (6) calculates the SINR for the general received area.
(6)$$SINR_{m,BS}^{n}=\frac{G_{m,BS}^{n}P_{m,BS}^{n}}{\sum_{k\neq n,k=1}G_{m,BS}^{k}P_{m,BS}^{k}+\eta_{m}}$$

In (6), $G_{m,BS}^{n}\;P_{m,BS}^{n}$ are calculated by multiplying $G_{m,BS}^{n}$, a propagation gain between BS-type RSU n and VUE m, and transmission power $P_{m,BS}^{n}$ transmitted from BS-type RSU n and VUE m. $G_{m,BS}^{k}$ and $P_{m,BS}^{k}$ represent the total interference received from BS-type RSUs in all adjacent cells except the supported RSUs k, where *η* is the thermal noise per RB. UE-type RSU also calculates the generated incoming SINR by (7).
(7)$$SINR_{m,UE}^{n}=\frac{G_{m,UE}^{n}P_{m,UE}^{n}}{\sum_{k\neq n,k=1}G_{m,UE}^{k}P_{m,UE}^{k}+\eta_{m}}$$

The UE type RSU is calculated by multiplying the propagation gain $G_{m,UE}^{n}$ and transmission power $P_{m,UE}^{n}$ between VUE m and UE type RSU n, as described above. In cases of interference, the SINR value is calculated by considering the interference from all UE type RSUs k in the simulation except for the UE-type RSU being serviced, and the interference between VUE m and these RSUs. The VUE-received SINR, obtained by applying the SINR calculated through each equation to the SLS, is shown in Figure 3. In the case of Figure 3a, a SINR map represents the SINR received when the VUE receives services from the BS-type RSU, and in the case of Figure 3b, the SINR received when the VUE receives services from the UE-type. We apply (8) to minimize the shaded area and select the supported RSU according to the SINR received from the RSU.
(8)$$SINR_{BStype}<SINR_{UEtype}\;for\;Serving\;BS\;type\;RSU\;SINR_{BStype}>SINR_{UEtype}\;for\;Serving\;UE\;type\;RSU$$

Figure 3c shows the result through the SINR map when using the proposed hybrid RSU method. As a result of the simulation, we confirm that the proposed hybrid RSU method reduces the shadow area by mitigating co-channel interference.

### 4.2. Rellaibilty Evaluation Based on BS-Type, UE-Type, and Hybrid RSU

We conduct a performance evaluation to confirm the reduction of the shaded area through the hybrid RSU method on the following three surfaces. Outage probability is defined as the probability that the received SINR of a VUE is less than the SINR threshold value presented, and the SINR threshold is set as the SINR value of the VUE when 5% of all VUEs in the simulation experience an outage. In addition, we evaluate and calculate the outage probability in each case using (9).
(9)$$P_{out}=1-P_{b}\left(SINR\geq SINR_{Threshold}\right)$$

In the case of Figure 4a, the cumulative distribution function (CDF) represents the received SINR values of VUEs that occur in the cases of UE-type RSU, BS-type RSU, and Hybrid RSU, and the SINR threshold value for checking the outage probability is set assuming that 5% of all VUEs experience an outage in the presented scenario. It can be observed that the received SINR values in the presented overall scenario are generally higher in the Hybrid RSU approach, and the SINR threshold values at which the outage probability can occur in each case are −4.85 dB for UE-type RSU, −2.64 dB for BS-type RSU, and −0.84 dB for Hybrid RSU, respectively. The presented Hybrid RSU has a higher SINR threshold value than UE/BS-type RSU, and overall, the received SINR is improved when the UE/BS-type RSUs cooperate. We confirm outage probability for each type by setting the SINR threshold value as the reference for the presented Hybrid RSU type to perform a more detailed evaluation and analysis. When we check outage probability at the presented SINR threshold value for UE/BS-type RSU, we find that 16% of all VUEs experienced an outage. When we apply the presented Hybrid RSU approach, the proportion of all VUEs experiencing an outage is reduced to 5% as the SINR of VUEs effectively increases, and the outage probability is reduced up to 11% compared to the UE/BS-type RSU approach. This indicates that the Hybrid RSU approach is helpful in the system. The decrease in the outage probability in the simulation can be represented as a reduction in the shadow area, and this approach is more efficient for resource allocation. Furthermore, the improvement in throughput occurs when the outage experienced by VUEs is reduced, which can be clearly observed in Figure 4c. Additionally, the evaluated VUE received interference refers to all interference received from adjacent cells, not the serving cell providing service to the VUE. The interference is calculated using Equation (10).
(10)$$\underset{k\neq n,k=1}{\sum }G_{m,UE}^{k}P_{m,UE}^{k}+\eta_{m}$$

In the case of interference, the UE-type RSU has a higher interference than the BS-type RSU, because of the denser distribution of UE-type RSUs than the BS type. Accordingly, the interference from other UE-type RSUs increases and the VUE receives a signal from a nearby UE-type RSU. Thus, we confirm a significant VUE interference due to the application of different parameters from the BS-type RSU. In the case of Figure 4b, it is an evaluation of the 50% interval of the cumulative distribution function (CDF) to confirm interference. It has a value of −77.37 dB for BS-type RSU, −50.71 dB for UE-type RSU, and −62.26 dB for hybrid RSU. In the case of throughput, we evaluate the 50% CDF and calculate throughput using (11).
(11)$$Throughput=\frac{TB}{Accounted_{TTIs}\times TTI_{Time}}$$

Throughput means the transport block (TB) size received per unit of time. The TB value represents the total number of bits received in the VUE, and TTI_Time_ represents the size of the maximum transmission time interval(TTI), and Accounted_TTIs_ represents the number of RBs assigned to the VUE receiver. Figure 4c shows the result of the VUE average throughput, which is 1.528 Mbp/s when the vehicle receives data from the BS-type RSU only, and 2.5 Mbp/s when receiving data from the UE-type RSU. The average throughput using the hybrid RSU method, which is the proposed BS/UE-type RSU cooperation scheme, is 3.06 Mbp/s, which shows a 100% performance improvement over 1.532 Mbp/s when data transmits from the BS-type RSU. In addition, we confirm that the average throughput is increased by 0.56 Mbp/s and the performance is improved by about 22% compared to the case of receiving data from the UE-type RSU. This result confirms that the proposed hybrid RSU method improves the reliability of vehicle users by mitigating the same interference channel and enhancing the quality of the communication system. Additionally, in a 5G-NR V2X environment, it is observed that the shadow area is minimized by applying the hybrid RSU method, and the average processing capacity of the vehicles is maximized through collaborative access between BS- and UE-type RSUs.

## 5. Improved V2I/N Link Reliability with Proposed Collaborative RSU for Resource Management

Each RSU configuration can apply various technologies through different and heterogeneous parameters. Therefore, mutual cooperation between RSUs can maximize throughput performance using the proposed methods. Each applicable technology uses dynamic inter-cell interface coordination (ICIC), coordinated scheduling coordinated multi-point (CS-CoMP), and 3D beamforming technologies for cell edge users in BS-type RSUs. In addition, performance improvement is confirmed by applying cell range expansion (CRE) technology at the UE-type RSUs, and analyzes the characteristics of each technology application.

### 5.1. BS-Type RSU with Resource Co-Operation Management

Quality of service (QoS) required in-vehicle communication. A resource cooperation management scheme improves the reliability of the entire communication system, by mitigating CCI in a cellular V2I network where BS-type RSUs and VUE communicate in a 5G-NR-V2X environment, and coordinated interference management is possible through dynamic ICIC and CS-CoMP technologies. For dynamic ICIC, we reallocate resources by redistributing frequencies using the fractional frequency reuse (FFR) method, which uses frequency reuse coefficients for cell edge users. Cell-centered users with multiple vehicles at a fixed full frequency are assigned a full-reuse (FR) spectrum, and users at the edge of the cell are assigned a frequency divided by a partial-reuse (PR) spectrum. The proposed dynamic ICIC allows allocating more resources to vehicles located at the edge of the cell, including intersections where we find many vehicles distributed. In this case, we set the SINR threshold by defining the criterion for dividing the PR band and FR band as 5% of the total users for allocating frequencies by distinguishing cell-center users and edge users. This approach allows us to determine whether to assign BBW. Then, using (12), we determine users assigned to the PR and FR bands, and apply dynamic ICIC technology.
(12)$$SINR_{vue}<SINR_{threshold}\;for\;PR\;zone\;users\;SINR_{vue}>SINR_{threshold}\;for\;FR\;zone\;users$$

In addition, applying CS-CoMP, with resource cooperation management can enhance the spectrum efficiency of users at the edge of a cell from signals received from other cell sites. It also improves received signal quality through coordination between signals transmitted from sites. To reduce unwanted CCI between BS-type RSUs, we apply a blocking-based scheduling process and each BS-type RSU receives feedback from the VUE through the central scheduler, and determines whether to use CS-CoMP. In addition, the VUE blocks interference signals from neighboring BSs with the same PRB, implements inter-sector muting, and applies CS-CoMP.

### 5.2. UE-type RSU with Cell Range Expansion

Cell range expansion technology supports higher spectral efficiency and energy efficiency in Heterogenic Networks (HetNet) consisting of conventional macro and small cells. We propose CRE technology to adjust the range of small-cell RSUs. CRE technology adds bias to the measured reception signal when connecting cell support VUE in HetNet, where macro cell RSUs and small-cell RSUs exist simultaneously, balancing the macro and small-cell RSUs [29]. A CRE technique would efficiently utilize UE-type RSUs with higher throughput performance in urban environments of 5G-NR-V2X networks, and prevent terminals adjacent to UE-type RSU boundaries from connecting to BS-type RSUs, expanding UE-type RSUs. When the received signal bias for the cell region expansion is *α*, we apply CRE using (13).
(13)$$SINR_{BStype}<SINR_{UEtype}+\alpha \;\;for\;Serving\;BS\;type\;RSU\;SINR_{BStype}>SINR_{UEtype}+\alpha \;\;for\;Serving\;UE\;type\;RSU$$

Here, we calculate the bias *α* value applied to the received signal, as shown in Equation (14).
(14)$$\alpha =\frac{P_{Rx}\left(u\right)}{P_{int}^{UE}\left(u\right)+P_{int}^{BS}\left(u\right)+N_{0}}-\frac{P_{Rx}\left(u\right)}{P_{int}^{BS}\left(u\right)+N_{0}}$$

Thus, we calculate each item of *α* considering each case of the SINR of (13). When applying the cell expansion technique, we calculate the α value by excluding interference from the BS-type RSU in the UE-type RSU.
(15)$$\begin{array}{cc} SINR\left(u\right) & =\frac{P_{Rx}\left(u\right)}{P_{int}^{UE}\left(u\right)+N_{0}}\;\mathrm{for}\;\mathrm{UE}\;\mathrm{type}\;\mathrm{RSU}\;\mathrm{SINR}\\ & =\frac{P_{Rx}\left(u\right)}{P_{int}^{BS}\left(u\right)+N_{0}}\;\mathrm{for}\;\mathrm{BS}\;\mathrm{type}\;\mathrm{RSU}\;\mathrm{SINR}\\ & =\frac{P_{Rx}\left(u\right)}{P_{int}^{UE}\left(u\right)+P_{int}^{BS}\left(u\right)+N_{0}}\;\mathrm{for}\;\mathrm{Cell}\;\mathrm{Range}\;\mathrm{Expansion} \end{array}$$

Here, $P_{Rx}\left(u\right)$ represents the received signal power of the VUE, and we calculate the total interference size received from the BS/UE-type RSU as $P_{int}^{BS}\left(u\right)$ and $P_{int}^{UE}\left(u\right)$ to model the SINR of the VUE.

### 5.3. BS-Type RSU with 3D Beamforming

In the case of 3D beamforming, combining the applied horizontal beam pattern adaptation with the vertical antenna pattern adaptation to maximize the desired user signal enhances the receiver’s SINR. Three-dimensional beamforming provides improved transmit power, spectral efficiency, and cell edge throughput. When integrated with ICIC, utilizing both horizontal and vertical planes in conjunction with 3D directional antenna patterns on the base station (BS) for inter-cell interference management, can lead to significant improvements in edge and peak throughput compared to conventional non-adaptive transmission, even at high latency/mobility values. This is because the degradation of channel state information (CSI) due to latency/mobility can be minimized. [30]. It avoids interference by allocating orthogonal resources for VUEs located at cell boundaries and connects to different BS-type RSUs. Configuring the antenna of 3D beamforming involves using the technology introduced in Section 3.2. In addition, we define the 3D antenna pattern by a pair of vertical and horizontal angles $\left(\theta^{\prime \prime },\emptyset^{\prime \prime }\right)$. The vertical and horizontal angle setting is an important parameter that enables the communication between distant nodes, by determining the power transmission location of the antenna in scenario using a directional transmission. The combination of vertical and horizontal patterns follows (16) [31,32].
(16)$$A_{dB}^{\prime \prime }\left(\theta^{\prime \prime },\emptyset^{\prime \prime }\right)=-\min\left\{-\left(A_{dB}^{\prime \prime }\left(\theta^{\prime \prime },\emptyset^{\prime \prime }=0{}^{\circ}\right)+A_{dB}^{\prime \prime }\left(\theta^{\prime \prime }=90{}^{\circ},\emptyset^{\prime \prime }\right)\right),A_{max}\right\}$$

The antenna element gain pattern consists of vertical pattern, A_dB_″ ($\theta^{\prime \prime }$, $\emptyset^{\prime \prime }$ = 0°), and horizontal pattern, A_dB_″($\theta^{\prime \prime }$ = 90°, $\emptyset^{\prime \prime }$), and shows the maximum gain and high directivity in the main lobe direction of 8 dBi of the single element radiation pattern. Equations (17) and (18) represent the vertical and horizontal patterns.
(17)$$A_{dB}^{\prime \prime }\left(\theta^{\prime \prime },\emptyset^{\prime \prime }=0{}^{\circ}\right)=-\min\left\{12{\left(\frac{\theta -90{}^{\circ}}{\theta_{3dB}}\right)}^{2},SLA_{V}\right\}$$
(18)$$A_{dB}^{\prime \prime }\left(\theta^{\prime \prime }=90{}^{\circ},\emptyset^{\prime \prime }\right)=-\min\left\{12{\left(\frac{\emptyset }{\emptyset_{3dB}}\right)}^{2},A_{max}\right\}$$

In the equation, θ_3dB_ and ∅_3dB_ mean vertical and horizontal 3 dB bandwidths, *SLA_V_* means the sidelobe level limit, and A_max_ indicates the mean maximum attenuation values. In the case of the vertical pattern, conditions corresponding to θ_3dB_ = 65°, *SLA_V_* = 30 dB, Further, $\theta^{\prime \prime }$ ∈ [0°, 180°] are satisfied, and in the case of the horizontal pattern, conditions ∅_3dB_ = 65°, *A_max_* = 30 dB, ∅ ∈ [180°, 180°] are satisfied.

To verify the antenna gain of 3D beamforming, a comparison is made between a single antenna configuration and an antenna array configuration consisting of a uniform rectangular panel array with rows and columns of (8, 8) elements. According to [33], using a single rectangular antenna panel can improve cellular capacity through vertical beamforming, and although the single-panel approach may sacrifice some diversity gain, it enables the activation of high-altitude (vertical) beamforming, increasing cellular capacity. MATLAB-based simulators are used to apply beamforming proposed in the 5G NR standard in accordance with the 3GPP document [25], and the resulting 3D antenna pattern, using the antenna parameters shown in Figure 5, demonstrates an additional gain of 18 dBi. The benefits of the 3D beamforming calculated through the analysis can be further enhanced by applying it to RSUs of the proposed hybrid RSU scheme, in conjunction with resource management techniques, such as dynamic ICIC and CS-CoMP, as previously suggested.

### 5.4. Simulation Results

In this study, we evaluate the proposed techniques to maximize the throughput performance of the hybrid RSU method. For the evaluation, we use MATLAB-based SLS to check and analyze throughput performance. The main SLS parameters apply the values shown in Table 2.

In the case of Figure 6a, the result of the BS/UE/Hybrid RSU throughput curve confirms the throughput performance when applying dynamic ICIC and CS-CoMP, which are resource cooperation management schemes, to a single BS-type RSU. It compares/verifies the throughput using the resource cooperation management technology with the hybrid RSU method. The dotted line graph represents the value before applying a specific technique, and the solid line graph represents the throughput curve after employing resource cooperation management. According to the graph, when applying (dynamic ICIC + CS-CoMP) to a single BS-type RSU, throughput improved by about 10%, from 1.528 [Mb/s] to 1.68 [Mb/s]. When applying the resource cooperation management scheme to the hybrid RSU method, the throughput performance improved by 24%, from 3.02 [Mb/s] to 3.768 [Mb/s].

In the case of Figure 6b, we compare the performance by applying the CRE technique to the UE-type RSU. The performance is simultaneously compared by application together with the previously proposed resource cooperation management technique. When checking the throughput at 50% CDF, which is an evaluation when applying CRE in the hybrid RSU method, throughput increases from 3.02 [Mb/s] to 4.246 [Mb/s], which corresponds to a 40% performance improvement. In addition, when applying CRE technology together with (dynamic ICIC + CS-CoMP) technology in hybrid RSU, throughput increases from 3.768 [Mb/s] to 4.537 [Mb/s], which corresponds to a performance improvement of about 20%. However, as the CDF value increases, the throughput is similar to that of the case where CRE technology is not applied, which affects the performance improvement of users with a lower throughput, but not for VUE with a higher throughput.

In the case of Figure 6c, performance is compared after applying 3D beamforming to the previously applied technology, and 3D beamforming is applied to a single BS-type RSU, which can confirm that the throughput improves by 85%, increasing from 1.762 [Mb/s] to 3.276 [Mb/s]. When 3D beamforming applies with the hybrid RSU method and the proposed technologies dynamic ICIC, CS-CoMP, and CRE are all applied, the throughput increases from 4.54 [Mb/s] to 5.85 [Mb/s], showing a 28% performance improvement. Table 3 shows the throughput for comparative performance improvement analysis when the proposed technology is applied to the hybrid RSU method. Throughput improves when we use each technology with the hybrid RSU method. In addition, when dynamic ICIC, CS-CoMP, and CRE technologies are applied at the same time, the throughput is improved by about 93%, from 3.02 [Mb/s] to 5.85 [Mb/s], compared to the hybrid RSU method without any technology applied, confirming improved reliability of the vehicle terminal. Simulation results show that the average throughput received by the vehicle in the simulator increases by about 282%, from 1.528 [Mb/s] to 5.85 [Mb/s], when transmitting and receiving V2X messages using only a single BS-type RSU and when applying all the proposed techniques. Compare to the case of transmitting and receiving V2X messages using only UE-type RSU as a single unit, it increases by about 133%, from 2.507 [Mb/s] to 5.85 [Mb/s].

## 6. Conclusions

In this paper, we propose a hybrid RSU method, a BS/UE-type RSU-based mutual cooperation scheme, to minimize the shaded areas under the 5G-NR V2X urban environments and improve communication reliability, to maximize traffic safety and road efficiency. To verify the performance of the proposed hybrid RSU method, we implement SLS according to the 3GPP technical documents for V2X. We analyze its performance using BS- and UE-type RSUs, respectively, within hybrid RSU scenarios. The proposed approach satisfies the necessary resource allocation requirements in the 5G-V2X environment, while minimizing outage issues caused by high vehicular mobility. It ensures reliability for all V2I/N links and maximizes the total ergodic throughput of the links. In addition, we evaluate performances when applying a resource cooperative management scheme, CRE, and 3D beamforming technologies, along with parameter analysis by each RSU type, to improve V2I/N link reliability. We check the outage probability, interference, and throughput for performance evaluation, to verify the reduced shaded areas and improve communication reliability. Simulation results show that a trade-off relationship occurs between interference and throughput when using independent RSUs. Applying the proposed hybrid RSU method results in a lower interference and higher throughput, thereby improving the reliability of V2X communication links. When dynamic ICIC, CS-CoMP, and 3D beamforming technologies are applied to BS-type RSUs to maximize performance, as the effective resource allocation of cell edge users is ensured through the relaxation of CCI between cells and the combination of horizontal and vertical beam patterns. We apply CRE to UE-type RSUs, and the average throughput improves by solving the problem of reception interference of UE-type RSUs due to the presence of high-power BS-type RSUs. As a result, we validate the significant performance improvement in throughput of the proposed hybrid RSU method and the proposed techniques in this paper, contributing to the overall reliability improvement of V2X communication links.

## Figures and Tables

**Figure 1 sensors-23-03989-f001:**
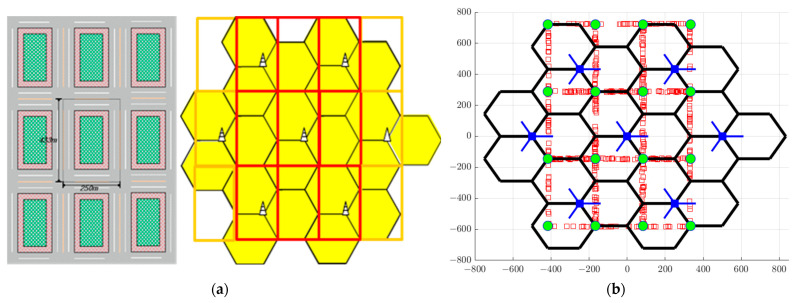
(**a**) Manhattan urban environment road and wrap−around model. (b) Network layout and vehicle distribution in a 5G−NR−V2X environment.

**Figure 2 sensors-23-03989-f002:**
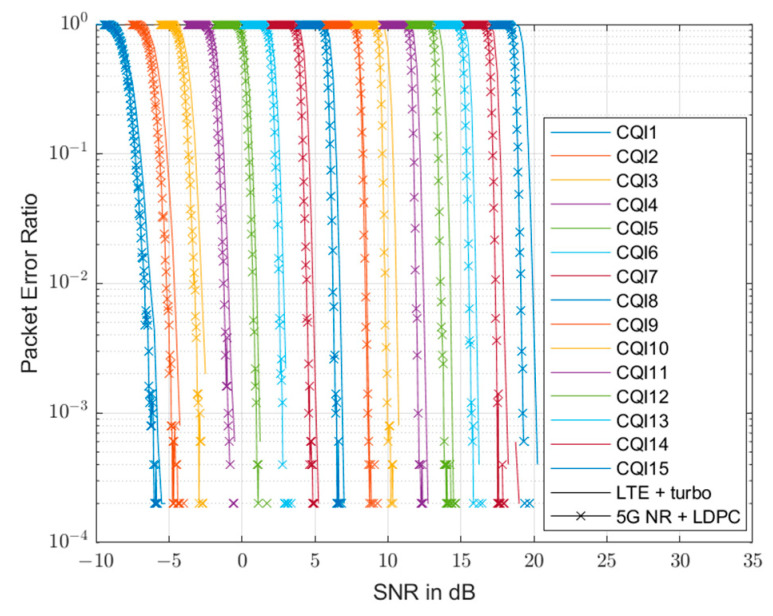
SNR vs. PER curve comparing LLS results when using turbo coding in the LTE environment, and LLS results when using LDPC coding in the 5G−NR environment.

**Figure 3 sensors-23-03989-f003:**
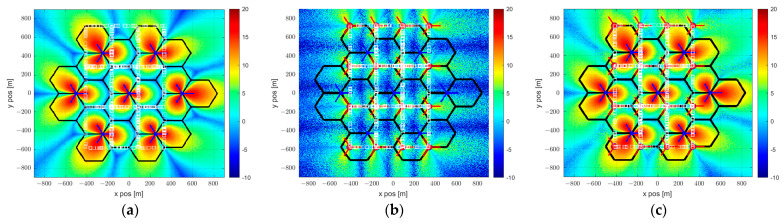
(**a**) BS−type RSU; (**b**) UE−type RSU; (**c**) Hybrid RSU SINR map.

**Figure 4 sensors-23-03989-f004:**
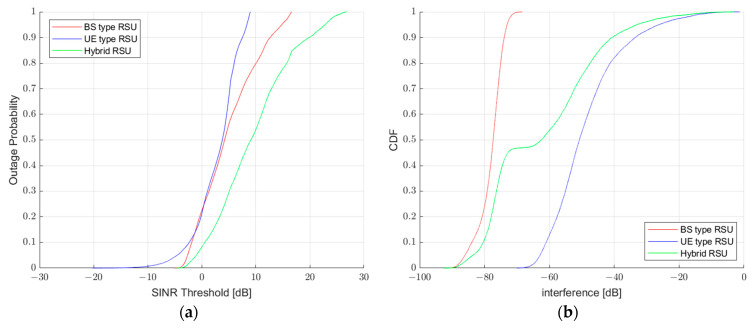
(**a**) 5G−NR V2X BS/UE/Hybrid RSU outage probability; (**b**) 5G−NR V2X BS/UE/Hybrid RSU interference; (**c**) 5G−NR V2X BS/UE/Hybrid RSU throughput.

**Figure 5 sensors-23-03989-f005:**
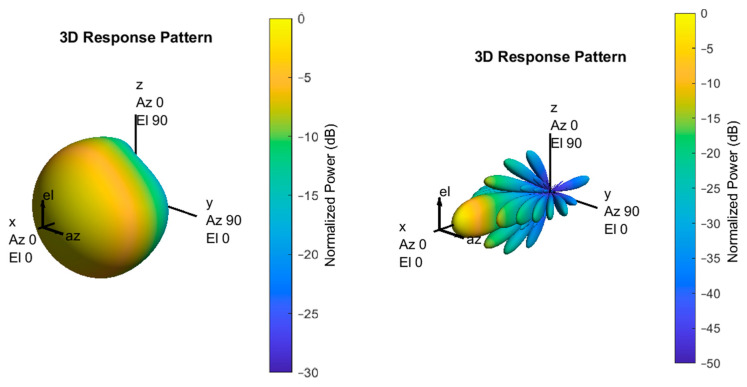
Three−dimensional beamforming antenna pattern.

**Figure 6 sensors-23-03989-f006:**
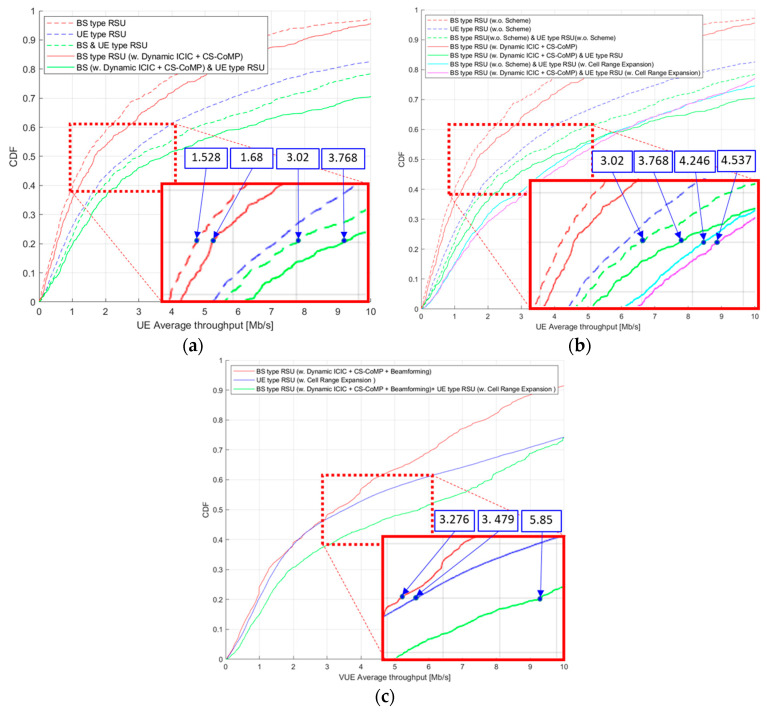
(**a**) VUE throughput applying dynamic ICIC + CS CoMP; (**b**) VUE throughput applying cell range expansion; (**c**) VUE throughput applying 3D beamforming.

**Table 1 sensors-23-03989-t001:** BS/UE-type RSU channel model parameters.

Parameter	5G-V2X (BS-Type RSU)	5G-V2X (UE-Type RSU)
Pathloss model	128.1 + 37.6log10(R), (R in kilometers)	WINNER + B1 Manhattan grid layout
Penetration loss	0 dB	0 dB
Shadowing	Log-normal Distribution Mean: 0 dB St. Dev.: 8 dB	Log-normal Distribution Mean: 0 dB St. Dev.: 3 dB
Decorrelation distance	50 m	10 m
Fading	3GPP Spatial Channel Model (SCM) NLOS	ITU-R Urban Micro-Clustered Delay Line Models (CDL) NLOS

**Table 2 sensors-23-03989-t002:** Simulation parameters.

Parameter	5G-V2X (BS-Type RSU)	5G-V2X (UE-Type RSU)
Carrier Freq.	6 GHz	6 GHz
Bandwidth	10 MHz	10 MHz
Subcarrier Spacing	15 kHz	15 kHz
RB Bandwidth	180 kHz	180 kHz
No. of PRBs	50	50
RSU Deployment	Hexagonal cells with 500 m ISD	Center of intersection
Noise Spectral Density	−174 dBm/Hz	−174 dBm/Hz
Tx Power	49 dBm	23 dBm
Max. Antenna Gain	15 dBi	5 dBi
Antenna Height	25 m	5 m
Antenna Element Pattern	For Macro BS	For UE-type RSU
Traffic Model	CAM	CAM
Scheduler	Proportional Fair	Proportional Fair

**Table 3 sensors-23-03989-t003:** Comparative analysis of throughput with application of various schemes.

Scheme	Dynamic ICIC + CS-CoMP	CRE	3D Beamforming	Throughput [Mb/s]
Hybrid RSU	X	X	X	3.02
	O	X	X	3.768
	O	O	X	4.537
	O	O	O	5.85

## Data Availability

Not applicable.

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
