# Peer review of "Resource Management for Collaborative 5G-NR-V2X RSUs to Enhance V2I/N Link Reliability"

_sensors, 2023, doi:10.3390/s23083989_

Round 1

Reviewer 1 Report

The authors propose an approach to ensure strict quality of service requirements for V2X communication using resource cooperation management and 3D beamforming technologies. While the paper has some valuable contributions, there are several points for improvement.

- the authors should clearly explain the added value of their approach compared to existing studies in the state of the art.

- the authors used a simulation tool, but they did not provide sufficient details about the simulator used. Additionally, the simulation only considered a single density and speed for all vehicles, which is not representative of the real-world scenario where vehicles can have different speeds.

- all parameters of equations (e.g., eq. 8, 9) need to be clearly defined. Furthermore, there are some errors in the references, such as references 17, 18, 19, and 20 not being found, and reference 24 being incorrect as the journal is Electronic Engineering and Systems.

- while the authors evaluated the impact of beamforming antennas, the results are not clear, and it does not present any new innovation.

Overall, the content of the paper is insufficient for publication and requires significant improvement. 

Reviewer 2 Report

The authors consider BS/UE-type RSU-based cooperation schemes aimed at minimizing shaded areas under the 5G-NR V2X urban environments and improving communication reliability to maximize traffic safety and road efficiency. In general, the paper is well structured, and presents a timely and valuable contribution in the field. However, there are several concerns (including the limited contribution, as well as the simple performance evaluation) that limit the strength of the submitted work. Specifically, I would raise the following comments.

1.       Only simulation results are presented in the paper with the use of a well-known tool, which limits the contribution of the paper. I would suggest the authors enriching the paper with the detailed description of the developed algorithms, analytical framework, if any.

2.       The authors say that the simulation’s VUE are distributed on the road according to the spatial Poisson process. However, in pure Poisson process, there is no minimum separation distance between the points, which is generally unneglectable property of V2X communications. What exact process is employed?

3.       The numerical results present mainly the throughput of V2X communication for the observed schemes under the fixed environment conditions. However, it could be worth providing the analysis of the considered metrics for various vehicle speeds, BS and VUE densities, antenna configurations, etc.

4.       There are some typos in the paper, e.g. small letter at the beginning of the paragraph (line 357), formula misprints (line 396-398).

Reviewer 3 Report

1. The article does not indicate the effect of the number of on-borde units (OBU) on signal propagation when using Hybrid RSU.

2. The article does not specify the installation altitude for the road unit (RSU) in the simulation.

Round 2

Reviewer 1 Report

The authors took into account all remarks.

However, the introduction can be improved in order to focus on the personal contribution.

Reviewer 2 Report

The authors have satisfactorily addressed my previous comments. The addition of new content has added to the overall value of the manuscript. Still, I would suggest to elaborate on the numerical analysis of outages, as the time in outage conditions is of no less importance then the probability of such conditions.
